# SQIL: Imitation Learning via Reinforcement Learning with Sparse Rewards

**Siddharth Reddy, Anca D. Dragan, Sergey Levine**
Department of Electrical Engineering and Computer Science
University of California, Berkeley
{sgr,anca,svlevine}@berkeley.edu

## Abstract

Learning to imitate expert behavior from demonstrations can be challenging, especially in environments with high-dimensional, continuous observations and unknown dynamics. Supervised learning methods based on behavioral cloning (BC) suffer from distribution shift: because the agent greedily imitates demonstrated actions, it can drift away from demonstrated states due to error accumulation. Recent methods based on reinforcement learning (RL), such as inverse RL and generative adversarial imitation learning (GAIL), overcome this issue by training an RL agent to match the demonstrations over a long horizon. Since the true reward function for the task is unknown, these methods learn a reward function from the demonstrations, often using complex and brittle approximation techniques that involve adversarial training. We propose a simple alternative that still uses RL, but does not require learning a reward function. The key idea is to provide the agent with an incentive to match the demonstrations over a long horizon, by encouraging it to return to demonstrated states upon encountering new, out-of-distribution states. We accomplish this by giving the agent a constant reward of $r = +1$ for matching the demonstrated action in a demonstrated state, and a constant reward of $r = 0$ for all other behavior. Our method, which we call *soft Q imitation learning* (SQIL), can be implemented with a handful of minor modifications to any standard Q-learning or off-policy actor-critic algorithm. Theoretically, we show that SQIL can be interpreted as a regularized variant of BC that uses a sparsity prior to encourage long-horizon imitation. Empirically, we show that SQIL outperforms BC and achieves competitive results compared to GAIL, on a variety of image-based and low-dimensional tasks in Box2D, Atari, and MuJoCo. This paper is a proof of concept that illustrates how a simple imitation method based on RL with constant rewards can be as effective as more complex methods that use learned rewards.

## 1 Introduction

Many sequential decision-making problems can be tackled by imitation learning: an expert demonstrates near-optimal behavior to an agent, and the agent attempts to replicate that behavior in novel situations (Argall et al., 2009). This paper considers the problem of training an agent to imitate an expert, given expert action demonstrations and the ability to interact with the environment. The agent does not observe a reward signal or query the expert, and does not know the state transition dynamics.

Standard approaches based on behavioral cloning (BC) use supervised learning to greedily imitate demonstrated actions, without reasoning about the consequences of actions (Pomerleau, 1991). As a result, compounding errors cause the agent to drift away from the demonstrated states (Ross et al., 2011). The problem with BC is that, when the agent drifts and encounters out-of-distribution states, the agent does not know how to return to the demonstrated states. Recent methods based on inverse reinforcement learning (IRL) overcome this issue by training an RL agent not only to imitate demonstrated actions, but also to visit demonstrated states (Ng et al., 2000; Wulfmeier et al., 2015; Finn et al., 2016b; Fu et al., 2017). This is also the core idea behind generative adversarial imitation learning (GAIL) (Ho & Ermon, 2016), which implements IRL using generative adversarial

networks (Goodfellow et al., 2014; Finn et al., 2016a). Since the true reward function for the task is unknown, these methods construct a reward signal from the demonstrations through adversarial training, making them difficult to implement and use in practice (Kurach et al., 2018).

The main idea in this paper is that the effectiveness of adversarial imitation methods can be achieved by a much simpler approach that does not require adversarial training, or indeed learning a reward function at all. Intuitively, adversarial methods encourage long-horizon imitation by providing the agent with (1) an incentive to imitate the demonstrated actions in demonstrated states, and (2) an incentive to take actions that lead it back to demonstrated states when it encounters new, out-of-distribution states. One of the reasons why adversarial methods outperform greedy methods, such as BC, is that greedy methods only do (1), while adversarial methods do both (1) and (2). Our approach is intended to do both (1) and (2) without adversarial training, by using constant rewards instead of learned rewards. The key idea is that, instead of using a learned reward function to provide a reward signal to the agent, we can simply give the agent a constant reward of $r = +1$ for matching the demonstrated action in a demonstrated state, and a constant reward of $r = 0$ for all other behavior.

We motivate this approach theoretically, by showing that it implements a regularized variant of BC that learns long-horizon imitation by (a) imposing a sparsity prior on the reward function implied by the imitation policy, and (b) incorporating information about the state transition dynamics into the imitation policy. Intuitively, our method accomplishes (a) by training the agent using an extremely sparse reward function – +1 for demonstrations, 0 everywhere else – and accomplishes (b) by training the agent with RL instead of supervised learning.

We instantiate our approach with soft Q-learning (Haarnoja et al., 2017) by initializing the agent's experience replay buffer with expert demonstrations, setting the rewards to a constant $r = +1$ in the demonstration experiences, and setting rewards to a constant $r = 0$ in all of the new experiences the agent collects while interacting with the environment. Since soft Q-learning is an off-policy algorithm, the agent does not necessarily have to visit the demonstrated states in order to experience positive rewards. Instead, the agent replays the demonstrations that were initially added to its buffer. Thus, our method can be applied in environments with stochastic dynamics and continuous states, where the demonstrated states are not necessarily reachable by the agent. We call this method *soft Q imitation learning* (SQIL).

The main contribution of this paper is SQIL: a simple and general imitation learning algorithm that is effective in MDPs with high-dimensional, continuous observations and unknown dynamics. We run experiments in four image-based environments – Car Racing, Pong, Breakout, and Space Invaders – and three low-dimensional environments – Humanoid, HalfCheetah, and Lunar Lander – to compare SQIL to two prior methods: BC and GAIL. We find that SQIL outperforms BC and achieves competitive results compared to GAIL. Our experiments illustrate two key benefits of SQIL: (1) that it can overcome the state distribution shift problem of BC without adversarial training or learning a reward function, which makes it easier to use, e.g., with images, and (2) that it is simple to implement using existing Q-learning or off-policy actor-critic algorithms.

## 2   SOFT Q IMITATION LEARNING

SQIL performs soft Q-learning (Haarnoja et al., 2017) with three small, but important, modifications: (1) it initially fills the agent's experience replay buffer with demonstrations, where the rewards are set to a constant $r = +1$; (2) as the agent interacts with the world and accumulates new experiences, it adds them to the replay buffer, and sets the rewards for these new experiences to a constant $r = 0$; and (3) it balances the number of demonstration experiences and new experiences (50% each) in each sample from the replay buffer.[1] These three modifications are motivated theoretically in Section 3, via an equivalence to a regularized variant of BC. Intuitively, these modifications create a simple reward structure that gives the agent an incentive to imitate the expert in demonstrated states, and to take actions that lead it back to demonstrated states when it strays from the demonstrations.

---

[1] SQIL resembles the Deep Q-learning from Demonstrations (DQfD) (Hester et al., 2017) and Normalized Actor-Critic (NAC) algorithms (Gao et al., 2018), in that it initially fills the agent's experience replay buffer with demonstrations. The key difference is that DQfD and NAC are RL algorithms that assume access to a reward signal, while SQIL is an imitation learning algorithm that does not require an extrinsic reward signal from the environment. Instead, SQIL automatically constructs a reward signal from the demonstrations.

---

**Algorithm 1** Soft Q Imitation Learning (SQIL)
1: Require $\lambda_{\text{samp}} \in \mathbb{R}_{\geq 0}, \mathcal{D}_{\text{demo}}$
2: Initialize $\mathcal{D}_{\text{samp}} \leftarrow \emptyset$
3: **while** $Q_{\boldsymbol{\theta}}$ not converged **do**
4:     $\boldsymbol{\theta} \leftarrow \boldsymbol{\theta} - \eta \nabla_{\boldsymbol{\theta}} (\delta^2(\mathcal{D}_{\text{demo}}, 1) + \lambda_{\text{samp}} \delta^2(\mathcal{D}_{\text{samp}}, 0))$ {See Equation 1}
5:     Sample transition $(s, a, s')$ with imitation policy $\pi(a|s) \propto \exp(Q_{\boldsymbol{\theta}}(s, a))$
6:     $\mathcal{D}_{\text{samp}} \leftarrow \mathcal{D}_{\text{samp}} \cup \{(s, a, s')\}$
7: **end while**

---

Crucially, since soft Q-learning is an off-policy algorithm, the agent does not necessarily have to visit the demonstrated states in order to experience positive rewards. Instead, the agent replays the demonstrations that were initially added to its buffer. Thus, SQIL can be used in stochastic environments with high-dimensional, continuous states, where the demonstrated states may never actually be encountered by the agent.

SQIL is summarized in Algorithm 1, where $Q_{\boldsymbol{\theta}}$ is the soft Q function, $\mathcal{D}_{\text{demo}}$ are demonstrations, $\delta^2$ is the squared soft Bellman error,

$$\delta^2(\mathcal{D}, r) \triangleq \frac{1}{|\mathcal{D}|} \sum_{(s,a,s') \in \mathcal{D}} \left( Q_{\boldsymbol{\theta}}(s, a) - \left( r + \gamma \log \left( \sum_{a' \in \mathcal{A}} \exp(Q_{\boldsymbol{\theta}}(s', a')) \right) \right) \right)^2, \quad (1)$$

and $r \in \{0, 1\}$ is a constant reward.[2] The experiments in Section 4 use a convolutional neural network or multi-layer perceptron to model $Q_{\boldsymbol{\theta}}$, where $\boldsymbol{\theta}$ are the weights of the neural network. Section A.3 in the appendix contains additional implementation details, including values for the hyperparameter $\lambda_{\text{samp}}$; note that the simple default value of $\lambda_{\text{samp}} = 1$ works well across a variety of environments.

As the imitation policy in line 5 of Algorithm 1 learns to behave more like the expert, a growing number of expert-like transitions get added to the buffer $\mathcal{D}_{\text{samp}}$ with an assigned reward of zero. This causes the effective reward for mimicking the expert to decay over time. Balancing the number of demonstration experiences and new experiences (50% each) sampled for the gradient step in line 4 ensures that this effective reward remains at least $1/(1 + \lambda_{\text{samp}})$, instead of decaying to zero. In practice, we find that this reward decay does not degrade performance if SQIL is halted once the squared soft Bellman error loss converges to a minimum (e.g., see Figure 8 in the appendix). Note that prior methods also require similar techniques: both GAIL and adversarial IRL (AIRL) (Fu et al., 2017) balance the number of positive and negative examples in the training set of the discriminator, and AIRL tends to require early stopping to avoid overfitting.

## 3 Interpreting SQIL as Regularized Behavioral Cloning

To understand why SQIL works, we sketch a surprising theoretical result: SQIL is equivalent to a variant of behavioral cloning (BC) that uses regularization to overcome state distribution shift.

BC is a simple approach that seeks to imitate the expert's actions using supervised learning – in particular, greedily maximizing the conditional likelihood of the demonstrated actions given the demonstrated states, without reasoning about the consequences of actions. Thus, when the agent makes small mistakes and enters states that are slightly different from those in the demonstrations, the distribution mismatch between the states in the demonstrations and those actually encountered by the agent leads to compounding errors (Ross et al., 2011). We show that SQIL is equivalent to augmenting BC with a regularization term that incorporates information about the state transition dynamics into the imitation policy, and thus enables long-horizon imitation.

### 3.1 Preliminaries

**Maximum entropy model of expert behavior.** SQIL is built on soft Q-learning, which assumes that expert behavior follows the maximum entropy model (Ziebart et al., 2010; Levine, 2018). In

---

[2]Equation 1 assumes discrete actions, but SQIL can also be used with continuous actions, as shown in Section 4.3.

an infinite-horizon Markov Decision Process (MDP) with a continuous state space $\mathcal{S}$ and discrete action space $\mathcal{A}$,[3] the expert is assumed to follow a policy $\pi$ that maximizes reward $R(s, a)$. The policy $\pi$ forms a Boltzmann distribution over actions,

$$\pi(a|s) \triangleq \frac{\exp\left(Q(s,a)\right)}{\sum_{a' \in \mathcal{A}} \exp\left(Q(s,a')\right)}, \tag{2}$$

where $Q$ is the soft Q function. The soft Q values are a function of the rewards and dynamics, given by the soft Bellman equation,

$$Q(s, a) \triangleq R(s, a) + \gamma \mathbb{E}_{s'} \left[ \log \left( \sum_{a' \in \mathcal{A}} \exp\left(Q(s', a')\right) \right) \right]. \tag{3}$$

In our imitation setting, the rewards and dynamics are unknown. The expert generates a fixed set of demonstrations $\mathcal{D}_{\text{demo}}$, by rolling out their policy $\pi$ in the environment and generating state transitions $(s, a, s') \in \mathcal{D}_{\text{demo}}$.

**Behavioral cloning (BC).** Training an imitation policy with standard BC corresponds to fitting a parametric model $\pi_{\boldsymbol{\theta}}$ that minimizes the negative log-likelihood loss,

$$\ell_{\text{BC}}(\boldsymbol{\theta}) \triangleq \sum_{(s,a) \in \mathcal{D}_{\text{demo}}} -\log \pi_{\boldsymbol{\theta}}(a|s). \tag{4}$$

In our setting, instead of explicitly modeling the policy $\pi_{\boldsymbol{\theta}}$, we can represent the policy $\pi$ in terms of a soft Q function $Q_{\boldsymbol{\theta}}$ via Equation 2:

$$\pi(a|s) \triangleq \frac{\exp\left(Q_{\boldsymbol{\theta}}(s,a)\right)}{\sum_{a' \in \mathcal{A}} \exp\left(Q_{\boldsymbol{\theta}}(s,a')\right)}. \tag{5}$$

Using this representation of the policy, we can train $Q_{\boldsymbol{\theta}}$ via the maximum-likelihood objective in Equation 4:

$$\ell_{\text{BC}}(\boldsymbol{\theta}) \triangleq \sum_{(s,a) \in \mathcal{D}_{\text{demo}}} -\left( Q_{\boldsymbol{\theta}}(s, a) - \log \left( \sum_{a' \in \mathcal{A}} \exp\left(Q_{\boldsymbol{\theta}}(s, a')\right) \right) \right). \tag{6}$$

However, optimizing the BC loss in Equation 6 does not in general yield a valid soft Q function $Q_{\boldsymbol{\theta}}$ – i.e., a soft Q function that satisfies the soft Bellman equation (Equation 3) with respect to the dynamics and some reward function. The problem is that the BC loss does not incorporate any information about the dynamics into the learning objective, so $Q_{\boldsymbol{\theta}}$ learns to greedily assign high values to demonstrated actions, without considering the state transitions that occur as a consequence of actions. As a result, $Q_{\boldsymbol{\theta}}$ may output arbitrary values in states that are off-distribution from the demonstrations $\mathcal{D}_{\text{demo}}$.

In Section 3.2, we describe a regularized BC algorithm that adds constraints to ensure that $Q_{\boldsymbol{\theta}}$ is a valid soft Q function with respect to some implicitly-represented reward function, and further regularizes the implicit rewards with a sparsity prior. In Section 3.3, we show that this approach recovers an algorithm similar to SQIL.

## 3.2   REGULARIZED BEHAVIORAL CLONING

Under the maximum entropy model described in Section 3.1, expert behavior is driven by a reward function, a soft Q function that computes expected future returns, and a policy that takes actions with high soft Q values. In the previous section, we used these assumptions to represent the imitation policy in terms of a model of the soft Q function $Q_{\boldsymbol{\theta}}$ (Equation 5). In this section, we represent the reward function implicitly in terms of $Q_{\boldsymbol{\theta}}$, as shown in Equation 7. This allows us to derive SQIL as a variant of BC that imposes a sparsity prior on the implicitly-represented rewards.

**Sparsity regularization.** The issue with BC is that, when the agent encounters states that are out-of-distribution with respect to $\mathcal{D}_{\text{demo}}$, $Q_{\boldsymbol{\theta}}$ may output arbitrary values. One solution from prior work

---

[3]Assuming a discrete action space simplifies our analysis. SQIL can be applied to continuous control tasks using existing sampling methods (Haarnoja et al., 2017; 2018), as illustrated in Section 4.3.

(Piot et al., 2014) is to regularize $Q_{\boldsymbol{\theta}}$ with a sparsity prior on the implied rewards – in particular, a penalty on the magnitude of the rewards $\sum_{s \in \mathcal{S}, a \in \mathcal{A}} |R_q(s, a)|$ implied by $Q_{\boldsymbol{\theta}}$ via the soft Bellman equation (Equation 3), where

$$R_q(s, a) \triangleq Q_{\boldsymbol{\theta}}(s, a) - \gamma \mathbb{E}_{s'} \left[ \log \left( \sum_{a' \in \mathcal{A}} \exp \left( Q_{\boldsymbol{\theta}}(s', a') \right) \right) \right]. \tag{7}$$

Note that the reward function $R_q$ is not explicitly modeled in this method. Instead, we directly minimize the magnitude of the right-hand side of Equation 7, which is equivalent to minimizing $|R_q(s, a)|$.

The purpose of the penalty on $|R_q(s, a)|$ is two-fold: (1) it imposes a sparsity prior motivated by prior work (Piot et al., 2013), and (2) it incorporates information about the state transition dynamics into the imitation learning objective, since $R_q(s, a)$ is a function of an expectation over next state $s'$. (2) is critical for learning long-horizon behavior that imitates the demonstrations, instead of greedy maximization of the action likelihoods in standard BC. For details, see Piot et al. (2014).

**Approximations for continuous states.** Unlike the discrete environments tested in Piot et al. (2014), we assume the continuous state space $\mathcal{S}$ cannot be enumerated. Hence, we approximate the penalty $\sum_{s \in \mathcal{S}, a \in \mathcal{A}} |R_q(s, a)|$ by estimating it from samples: transitions $(s, a, s')$ observed in the demonstrations $\mathcal{D}_{\text{demo}}$, as well as additional rollouts $\mathcal{D}_{\text{samp}}$ periodically sampled during training using the latest imitation policy. This approximation, which follows the standard approach to constraint sampling (Calafiore & Dabbene, 2006), ensures that the penalty covers the state distribution actually encountered by the agent, instead of only the demonstrations.

To make the penalty continuously differentiable, we introduce an additional approximation: instead of penalizing the absolute value $|R_q(s, a)|$, we penalize the squared value $(R_q(s, a))^2$. Note that since the reward function $R_q$ is not explicitly modeled, but instead defined via $Q_{\boldsymbol{\theta}}$ in Equation 7, the squared penalty $(R_q(s, a))^2$ is equivalent to the squared soft Bellman error $\delta^2(\mathcal{D}_{\text{demo}} \cup \mathcal{D}_{\text{samp}}, 0)$ from Equation 1.

**Regularized BC algorithm.** Formally, we define the regularized BC loss function adapted from Piot et al. (2014) as

$$\ell_{\text{RBC}}(\boldsymbol{\theta}) \triangleq \ell_{\text{BC}}(\boldsymbol{\theta}) + \lambda \delta^2(\mathcal{D}_{\text{demo}} \cup \mathcal{D}_{\text{samp}}, 0), \tag{8}$$

where $\lambda \in \mathbb{R}_{\geq 0}$ is a constant hyperparameter, and $\delta^2$ denotes the squared soft Bellman error defined in Equation 1. The BC loss encourages $Q_{\boldsymbol{\theta}}$ to output high values for demonstrated actions at demonstrated states, and the penalty term propagates those high values to nearby states. In other words, $Q_{\boldsymbol{\theta}}$ outputs high values for actions that lead to states from which the demonstrated states are reachable. Hence, when the agent finds itself far from the demonstrated states, it takes actions that lead it back to the demonstrated states.

The RBC algorithm follows the same procedure as Algorithm 1, except that in line 4, RBC takes a gradient step on the RBC loss from Equation 8 instead of the SQIL loss.

## 3.3 CONNECTION BETWEEN SQIL AND REGULARIZED BEHAVIORAL CLONING

The gradient of the RBC loss in Equation 8 is proportional to the gradient of the SQIL loss in line 4 of Algorithm 1, plus an additional term that penalizes the soft value of the initial state $s_0$ (full derivation in Section A.1 of the appendix):

$$\nabla_{\boldsymbol{\theta}} \ell_{\text{RBC}}(\boldsymbol{\theta}) \propto \nabla_{\boldsymbol{\theta}} \left( \delta^2(\mathcal{D}_{\text{demo}}, 1) + \lambda_{\text{samp}} \delta^2(\mathcal{D}_{\text{samp}}, 0) + V(s_0) \right). \tag{9}$$

In other words, SQIL solves a similar optimization problem to RBC. The reward function in SQIL also has a clear connection to the sparsity prior in RBC: SQIL imposes the sparsity prior from RBC, by training the agent with an extremely sparse reward function $-r = +1$ at the demonstrations, and $r = 0$ everywhere else. Thus, SQIL can be motivated as a practical way to implement the ideas for regularizing BC proposed in Piot et al. (2014).

The main benefit of using SQIL instead of RBC is that SQIL is trivial to implement, since it only requires a few small changes to any standard Q-learning implementation (see Section 2). Extending SQIL to MDPs with a continuous action space is also easy, since we can simply replace Q-learning

with an off-policy actor-critic method (Haarnoja et al., 2018) (see Section 4.3). Given the difficulty of implementing deep RL algorithms correctly (Henderson et al., 2018), this flexibility makes SQIL more practical to use, since it can be built on top of existing implementations of deep RL algorithms. Furthermore, the ablation study in Section 4.4 suggests that SQIL actually performs better than RBC.

## 4 EXPERIMENTAL EVALUATION

Our experiments aim to compare SQIL to existing imitation learning methods on a variety of tasks with high-dimensional, continuous observations, such as images, and unknown dynamics. We benchmark SQIL against BC and GAIL[4] on four image-based games – Car Racing, Pong, Breakout, and Space Invaders – and three state-based tasks – Humanoid, HalfCheetah, and Lunar Lander (Brockman et al., 2016; Bellemare et al., 2013; Todorov et al., 2012). We also investigate which components of SQIL contribute most to its performance via an ablation study on the Lunar Lander game. Section A.3 in the appendix contains additional experimental details.

### 4.1 TESTING GENERALIZATION IN IMAGE-BASED CAR RACING

The goal of this experiment is to study not only how well each method can mimic the expert demonstrations, but also how well they can acquire policies that generalize to new states that are not seen in the demonstrations. To do so, we train the imitation agents in an environment with a different initial state distribution $\mathcal{S}_0^{\text{train}}$ than that of the expert demonstrations $\mathcal{S}_0^{\text{demo}}$, allowing us to systematically control the mismatch between the distribution of states in the demonstrations and the states actually encountered by the agent. We run experiments on the Car Racing game from OpenAI Gym. To create $\mathcal{S}_0^{\text{train}}$, the car is rotated 90 degrees so that it begins perpendicular to the track, instead of parallel to the track as in $\mathcal{S}_0^{\text{demo}}$. This intervention presents a significant generalization challenge to the imitation learner, since the expert demonstrations do not contain any examples of states where the car is perpendicular to the road, or even significantly off the road axis. The agent must learn to make a tight turn to get back on the road, then stabilize its orientation so that it is parallel to the road, and only then proceed forward to mimic the expert demonstrations.

The results in Figure 1 show that SQIL and BC perform equally well when there is no variation in the initial state. The task is easy enough that even BC achieves a high reward. Note that, in the unperturbed condition (right column), BC substantially outperforms GAIL, despite the well-known shortcomings of BC. This indicates that the adversarial optimization in GAIL can substantially hinder

|            | Domain Shift ($\mathcal{S}_0^{\text{train}}$) | No Shift ($\mathcal{S}_0^{\text{demo}}$) |
|------------|-------------------------------------------------|--------------------------------------------|
| Random     | $-21 \pm 56$                                    | $-68 \pm 4$                                 |
| BC         | $-45 \pm 18$                                    | $\mathbf{698 \pm 10}$                       |
| GAIL-DQL   | $-97 \pm 3$                                     | $-66 \pm 8$                                  |
| SQIL (Ours)| $\mathbf{375 \pm 19}$                           | $\mathbf{704 \pm 6}$                        |
| Expert     | $480 \pm 11$                                    | $704 \pm 79$                                |

Figure 1: Average reward on 100 episodes after training. Standard error on three random seeds.

learning, even in settings where standard BC is sufficient. SQIL performs much better than BC when starting from $\mathcal{S}_0^{\text{train}}$, showing that SQIL is capable of generalizing to a new initial state distribution, while BC is not. SQIL learns to make a tight turn that takes the car through the grass and back onto the road, then stabilizes the car's orientation so that it is parallel to the track, and then proceeds forward like the expert does in the demonstrations. BC tends to drive straight ahead into the grass instead of turning back onto the road.

---

[4] For all the image-based tasks, we implement a version of GAIL that uses deep Q-learning (GAIL-DQL) instead of TRPO as in the original GAIL paper (Ho & Ermon, 2016), since Q-learning performs better than TRPO in these environments, and because this allows for a head-to-head comparison of SQIL and GAIL: both algorithms use the same underlying RL algorithm, but provide the agent with different rewards – SQIL provides constant rewards, while GAIL provides learned rewards. We use the standard GAIL-TRPO method as a baseline for all the low-dimensional tasks, since TRPO performs better than Q-learning in these environments.

The original GAIL method implicitly encodes prior knowledge – namely, that terminating an episode is either always desirable or always undesirable. As pointed out in Kostrikov et al. (2019), this makes comparisons to alternative methods unfair. We implement the unbiased version of GAIL proposed by Kostrikov et al. (2019), and use this in all of the experiments. Comparisons to the biased version with implicit termination knowledge are included in Section A.2 in the appendix.

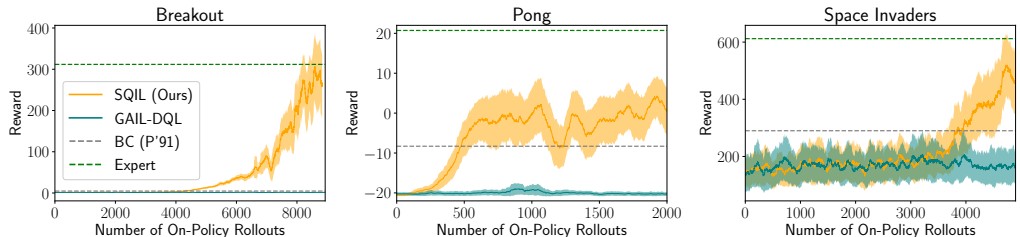

Figure 2: Image-based Atari. Smoothed with a rolling window of 100 episodes. Standard error on three random seeds. X-axis represents amount of interaction with the environment (not expert demonstrations).

SQIL outperforms GAIL in both conditions. Since SQIL and GAIL both use deep Q-learning for RL in this experiment, the gap between them may be attributed to the difference in the reward functions they use to train the agent. SQIL benefits from providing a constant reward that does not require fitting a discriminator, while GAIL struggles to train a discriminator to provide learned rewards directly from images.

## 4.2 IMAGE-BASED EXPERIMENTS ON ATARI

The results in Figure 2 show that SQIL outperforms BC on Pong, Breakout, and Space Invaders – additional evidence that BC suffers from compounding errors, while SQIL does not. SQIL also outperforms GAIL on all three games, illustrating the difficulty of using GAIL to train an image-based discriminator, as in Section 4.1.

## 4.3 INSTANTIATING SQIL FOR CONTINUOUS CONTROL IN LOW-DIMENSIONAL MUJOCO

The experiments in the previous sections evaluate SQIL on MDPs with a discrete action space. This section illustrates how SQIL can be adapted to continuous actions. We instantiate SQIL using soft actor-critic (SAC) – an off-policy RL algorithm that can solve continuous control tasks (Haarnoja et al., 2018). In particular, SAC is modified in the following ways: (1) the agent's experience replay buffer is initially filled with expert demonstrations, where rewards are set to $r = +1$, (2) when taking gradient steps to fit the agent's soft Q function, a balanced number of demonstration experiences and new experiences (50% each) are sampled from the replay buffer, and (3) the agent observes rewards of $r = 0$ during its interactions with the environment, instead of an extrinsic reward signal that specifies the desired task. This instantiation of SQIL is compared to GAIL on the Humanoid (17 DoF) and HalfCheetah (6 DoF) tasks from MuJoCo.

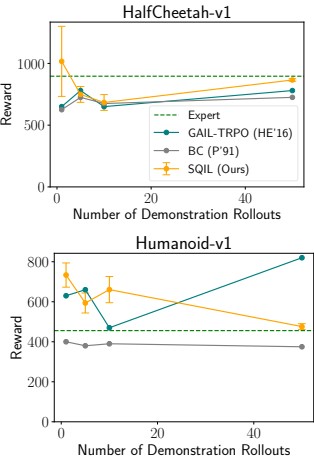

The results show that SQIL outperforms BC and performs comparably to GAIL on both tasks, demonstrating that SQIL can be successfully deployed on problems with continuous actions, and that SQIL can perform well even with a small number of demonstrations. This experiment also illustrates how SQIL can be run on top of SAC or any other off-policy value-based RL algorithm.

Figure 3: SQIL: best performance on 10 consecutive training episodes. BC, GAIL: results from Dhariwal et al. (2017).

## 4.4 ABLATION STUDY ON LOW-DIMENSIONAL LUNAR LANDER

We hypothesize that SQIL works well because it combines information about the expert's policy from demonstrations with information about the environment dynamics from rollouts of the imitation policy periodically sampled during training. We also expect RBC to perform comparably to SQIL, since their objectives are similar. To test these hypotheses, we conduct an ablation study using the Lunar Lander game from OpenAI Gym. As in Section 4.1, we control the mismatch between the

distribution of states in the demonstrations and the states encountered by the agent by manipulating the initial state distribution. To create $\mathcal{S}_0^{\text{train}}$, the agent is placed in a starting position never visited in the demonstrations.

In the first variant of SQIL, $\lambda_{\text{samp}}$ is set to zero, to prevent SQIL from using additional samples drawn from the environment (see line 4 of Algorithm 1). This comparison tests if SQIL really needs to interact with the environment, or if it can rely solely on the demonstrations. In the second condition, $\gamma$ is set to zero to prevent SQIL from accessing information about state transitions (see Equation 1 and line 4 of Algorithm 1). This comparison tests if SQIL is actually extracting information about the dynamics from the samples, or if it can perform just as well with a naïve regularizer (setting $\gamma$ to zero effectively imposes a penalty on the L2-norm of the soft Q values instead of the squared soft Bellman error). In the third condition, a uniform random policy is used to sample additional rollouts, instead of the imitation policy $\pi_\theta$ (see line 6 of Algorithm 1). This comparison tests how important it is that the samples cover the states encountered by the agent during training. In the fourth condition, we use RBC to optimize the loss in Equation 8. instead of using SQIL to optimize the loss in line 4 of Algorithm 1. This comparison tests the effect of the additional $V(s_0)$ term in RBC vs. SQIL (see Equation 9).

The results in Figure 4 show that all methods perform well when there is no variation in the initial state. When the initial state is varied, SQIL performs significantly better than BC, GAIL, and the ablated variants of SQIL. This confirms our hypothesis that SQIL needs to sample from the environment using the imitation policy, and relies on information about the dynamics encoded in the samples. Surprisingly, SQIL outperforms RBC by a large margin, suggesting that the penalty on the soft value of the initial state $V(s_0)$, which is present in RBC but not in SQIL (see Equation 9), degrades performance.

|  |  | Domain Shift ($\mathcal{S}_0^{\text{train}}$) | No Shift ($\mathcal{S}_0^{\text{demo}}$) |
|---|---|---|---|
|  | Random | $0.10 \pm 0.30$ | $0.04 \pm 0.02$ |
|  | BC | $0.07 \pm 0.03$ | $\mathbf{0.93 \pm 0.03}$ |
|  | GAIL-TRPO | $0.67 \pm 0.04$ | $\mathbf{0.93 \pm 0.03}$ |
|  | SQIL (Ours) | $\mathbf{0.89 \pm 0.02}$ | $0.88 \pm 0.03$ |
| Ablation | $\lambda_{\text{samp}} = 0$ | $0.12 \pm 0.02$ | $\mathbf{0.87 \pm 0.02}$ |
|  | $\gamma = 0$ | $0.41 \pm 0.02$ | $\mathbf{0.84 \pm 0.02}$ |
|  | $\pi = \text{Unif}$ | $0.47 \pm 0.02$ | $0.82 \pm 0.02$ |
|  | RBC | $0.66 \pm 0.02$ | $\mathbf{0.89 \pm 0.01}$ |
|  | Expert | $0.93 \pm 0.03$ | $0.89 \pm 0.31$ |

Figure 4: Best success rate on 100 consecutive episodes during training. Standard error on five random seeds. Performance bolded if at least within one standard error of expert.

## 5 DISCUSSION AND RELATED WORK

**Related work.** Concurrently with SQIL, two other imitation learning algorithms that use constant rewards instead of a learned reward function were developed (Sasaki et al., 2019; Wang et al., 2019). We see our paper as contributing additional evidence to support this core idea, rather than proposing a competing method. First, SQIL is derived from sparsity-regularized BC, while the prior methods are derived from an alternative formulation of the IRL objective (Sasaki et al., 2019) and from support estimation methods (Wang et al., 2019), showing that different theoretical approaches independently lead to using RL with constant rewards as an alternative to adversarial training – a sign that this idea may be a promising direction for future work. Second, SQIL is shown to outperform BC and GAIL in domains that were not evaluated in Sasaki et al. (2019) or Wang et al. (2019) – in particular, tasks with image observations and significant shift in the state distribution between the demonstrations and the training environment.

**Summary.** We contribute the SQIL algorithm: a general method for learning to imitate an expert given action demonstrations and access to the environment. Simulation experiments on tasks with high-dimensional, continuous observations and unknown dynamics show that our method outperforms BC and achieves competitive results compared to GAIL, while being simple to implement on top of existing off-policy RL algorithms.

**Limitations and future work.** We have not yet proven that SQIL matches the expert's state occupancy measure in the limit of infinite demonstrations. One direction for future work would be to rigorously show whether or not SQIL has this property. Another direction would be to extend SQIL to recover not just the expert's policy, but also their reward function; e.g., by using a parameterized reward function to model rewards in the soft Bellman error terms, instead of using constant rewards. This could provide a simpler alternative to existing adversarial IRL algorithms.

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

## A APPENDIX

### A.1 DERIVATION OF RBC GRADIENT

Let $\tau = (s_0, a_0, s_1, ..., s_T)$ denote a rollout, where $s_T$ is an absorbing state. Let $V$ denote the soft value function,

$$V(s) \triangleq \log\left(\sum_{a \in \mathcal{A}} \exp\left(Q_{\boldsymbol{\theta}}(s, a)\right)\right). \tag{10}$$

Splitting up the squared soft Bellman error terms for $\mathcal{D}_{\text{demo}}$ and $\mathcal{D}_{\text{samp}}$ in Equation 8,

$$\nabla \ell_{\text{RBC}}(\boldsymbol{\theta}) = \sum_{\tau \in \mathcal{D}_{\text{demo}}} \sum_{t=0}^{T-1} -(\nabla Q_{\boldsymbol{\theta}}(s_t, a_t) - \nabla V(s_t))$$

$$+ \lambda_{\text{demo}} \sum_{\tau \in \mathcal{D}_{\text{demo}}} \sum_{t=0}^{T-1} \nabla(Q_{\boldsymbol{\theta}}(s_t, a_t) - \gamma V(s_{t+1}))^2 + \lambda_{\text{samp}} \nabla \delta^2(\mathcal{D}_{\text{samp}}, 0)$$

$$= \sum_{\tau \in \mathcal{D}_{\text{demo}}} \sum_{t=0}^{T-1} \nabla V(s_t) - \gamma \nabla V(s_{t+1})$$

$$+ \lambda_{\text{demo}} \nabla \delta^2\left(\mathcal{D}_{\text{demo}}, \frac{1}{2\lambda_{\text{demo}}}\right) + \lambda_{\text{samp}} \nabla \delta^2(\mathcal{D}_{\text{samp}}, 0). \tag{11}$$

Setting $\gamma \triangleq 1$ turns the inner sum in the first term into a telescoping sum:

$$(11) = \sum_{\tau \in \mathcal{D}_{\text{demo}}} (\nabla V(s_0) - \nabla V(s_T)) + \lambda_{\text{demo}} \nabla \delta^2\left(\mathcal{D}_{\text{demo}}, \frac{1}{2\lambda_{\text{demo}}}\right) + \lambda_{\text{samp}} \nabla \delta^2(\mathcal{D}_{\text{samp}}, 0). \tag{12}$$

Since $s_T$ is assumed to be absorbing, $V(s_T)$ is zero. Thus,

$$(12) = \sum_{s_0 \in \mathcal{D}_{\text{demo}}} \nabla V(s_0) + \lambda_{\text{demo}} \nabla \delta^2\left(\mathcal{D}_{\text{demo}}, \frac{1}{2\lambda_{\text{demo}}}\right) + \lambda_{\text{samp}} \nabla \delta^2(\mathcal{D}_{\text{samp}}, 0), \tag{13}$$

In our experiments, we have that all the demonstration rollouts start at the same initial state $s_0$.[5] Thus,

$$(13) \propto \nabla\left(\delta^2(\mathcal{D}_{\text{demo}}, 1) + \lambda_{\text{samp}} \delta^2(\mathcal{D}_{\text{samp}}, 0) + V(s_0)\right), \tag{14}$$

where $\lambda_{\text{samp}} \in \mathbb{R}_{\geq 0}$ is a constant hyperparameter.

### A.2 COMPARING THE BIASED AND UNBIASED VARIANTS OF GAIL

As discussed in Section 4, to correct the original GAIL method's biased handling of rewards at absorbing states, we implement the suggested changes to GAIL in Section 4.2 of Kostrikov et al. (2019): adding a transition to an absorbing state and a self-loop at the absorbing state to the end of each rollout sampled from the environment, and adding a binary feature to the observations indicating whether or not a state is absorbing. This enables GAIL to learn a non-zero reward for absorbing states. We refer to the original, biased GAIL method as GAIL-DQL-B and GAIL-TRPO-B, and the unbiased version as GAIL-DQL-U and GAIL-TRPO-U.

The mechanism for learning terminal rewards proposed in Kostrikov et al. (2019) does not apply to SQIL, since SQIL does not learn a reward function. SQIL implicitly assumes a reward of zero at absorbing states in demonstrations. This is the case in all our experiments, which include some environments where terminating the episode is always undesirable (e.g., walking without falling down) and other environments where success requires terminating the episode (e.g., landing at a target), suggesting that SQIL is not sensitive to the choice of termination reward, and neither significantly benefits nor is significantly harmed by setting the termination reward to zero.

---

[5]Demonstration rollouts may still vary due to the stochasticity of the expert policy.

|  | Domain Shift ($\mathcal{S}_0^{\text{train}}$) | No Shift ($\mathcal{S}_0^{\text{demo}}$) |
|---|---|---|
| Random | $-21 \pm 56$ | $-68 \pm 4$ |
| BC | $-45 \pm 18$ | $\mathbf{698 \pm 10}$ |
| GAIL-DQL-B | $-91 \pm 4$ | $-34 \pm 21$ |
| GAIL-DQL-U | $-97 \pm 3$ | $-66 \pm 8$ |
| SQIL (Ours) | $\mathbf{375 \pm 19}$ | $\mathbf{704 \pm 6}$ |
| Expert | $480 \pm 11$ | $704 \pm 79$ |

Figure 5: Image-based Car Racing. Average reward on 100 episodes after training. Standard error on three random seeds.

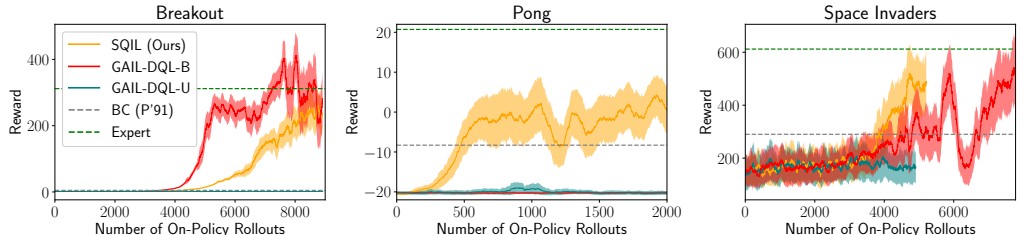

Figure 6: Image-based Atari. Smoothed with a rolling window of 100 episodes. Standard error on three random seeds. X-axis represents amount of interaction with the environment (not expert demonstrations).

**Car Racing.** The results in Figure 5 show that both the biased (GAIL-DQL-B) and unbiased (GAIL-DQL-U) versions of GAIL perform equally poorly. The problem of training an image-based discriminator for this task may be difficult enough that even with an unfair bias toward avoiding crashes that terminate the episode, GAIL-DQL-B does not perform better than GAIL-DQL-U.

**Atari.** The results in Figure 6 show that SQIL outperforms both variants of GAIL on Pong and the unbiased version of GAIL (GAIL-DQL-U) on Breakout and Space Invaders, but performs comparably to the biased version of GAIL (GAIL-DQL-B) on Space Invaders and worse than it on Breakout. This may be due to the fact that in Breakout and Space Invaders, the agent has multiple lives – five in Breakout, and three in Space Invaders – and receives a termination signal that the episode has ended after losing each life. Thus, the agent experiences many more episode terminations than in Pong, exacerbating the bias in the way the original GAIL method handles rewards at absorbing states. Our implementation of GAIL-DQL-B in this experiment provides a learned reward of $r(s, a) = -\log(1 - D(s, a))$, where $D$ is the discriminator (see Section A.3 in the appendix for details). The learned reward is always positive, while the implicit reward at an absorbing state is zero. Thus, the agent is inadvertently encouraged to avoid terminating the episode. For Breakout and Space Invaders, this just happens to be the right incentive, since the objective is to stay alive as long as possible. GAIL-DQL-B outperforms SQIL in Breakout and performs comparably to SQIL in Space Invaders because GAIL-DQL-B is accidentally biased in the right way.

**Lunar Lander.** The results in Figure 7 show that when the initial state is varied, SQIL outperforms the unbiased variant of GAIL (GAIL-TRPO-U), but underperforms against the biased version of GAIL (GAIL-TRPO-B). The latter result is likely due to the fact that the implementation of GAIL-TRPO-B we used in this experiment provides a learned reward of $r(s, a) = \log(D(s, a))$, where $D$ is the discriminator (see Section A.3 in the appendix for details). The learned reward is always negative, while the implicit reward at an absorbing state is zero. Thus, the agent is inadvertently encouraged to terminate the episode quickly. For the Lunar Lander game, this just happens to be the right incentive, since the objective is to land on the ground and thereby terminate the episode. As in the Atari experiments, GAIL-TRPO-B performs better than SQIL in this experiment because GAIL-TRPO-B is accidentally biased in the right way.

|  | | Domain Shift ($\mathcal{S}_0^{\text{train}}$) | No Shift ($\mathcal{S}_0^{\text{demo}}$) |
|---|---|---|---|
|  | Random | $0.10 \pm 0.30$ | $0.04 \pm 0.02$ |
|  | BC | $0.07 \pm 0.03$ | $\mathbf{0.93 \pm 0.03}$ |
|  | GAIL-TRPO-B (HE'16) | $\mathbf{0.98 \pm 0.01}$ | $\mathbf{0.95 \pm 0.02}$ |
|  | GAIL-TRPO-U | $0.67 \pm 0.04$ | $\mathbf{0.93 \pm 0.03}$ |
|  | SQIL (Ours) | $\mathbf{0.89 \pm 0.02}$ | $\mathbf{0.88 \pm 0.03}$ |
| Ablation | $\lambda_{\text{samp}} = 0$ | $0.12 \pm 0.02$ | $\mathbf{0.87 \pm 0.02}$ |
|  | $\gamma = 0$ | $0.41 \pm 0.02$ | $\mathbf{0.84 \pm 0.02}$ |
|  | $\pi = \text{Unif}$ | $0.47 \pm 0.02$ | $0.82 \pm 0.02$ |
|  | RBC | $0.66 \pm 0.02$ | $\mathbf{0.89 \pm 0.01}$ |
|  | Expert | $0.93 \pm 0.03$ | $0.89 \pm 0.31$ |

Figure 7: Low-dimensional Lunar Lander. Best success rate on 100 consecutive episodes during training. Standard error on five random seeds. Performance bolded if at least within one standard error of expert.

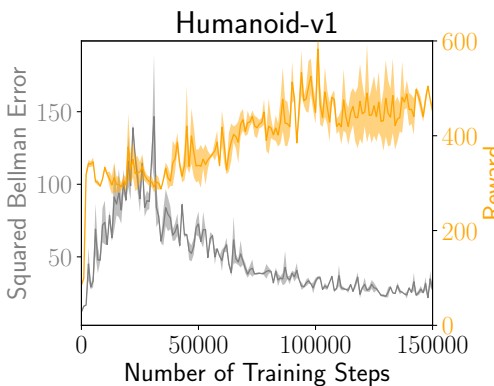

Figure 8: Standard error over two random seeds. No smoothing across training steps.

### A.3 IMPLEMENTATION DETAILS

To ensure fair comparisons, the same network architectures were used to evaluate SQIL, GAIL, and BC. For Lunar Lander, we used a network architecture with two fully-connected layers containing 128 hidden units each to represent the Q network in SQIL, the policy and discriminator networks in GAIL, and the policy network in BC. For Car Racing, we used four convolutional layers (following (Ha & Schmidhuber, 2018)) and two fully-connected layers containing 256 hidden units each. For Humanoid and HalfCheetah, we used two fully-connected layers containing 256 hidden units each. For Atari, we used the convolutional neural network described in (Mnih et al., 2015) to represent the Q network in SQIL, as well as the Q network and discriminator network in GAIL.

To ensure fair comparisons, the same demonstration data were used to train SQIL, GAIL, and BC. For Lunar Lander, we collected 100 demonstration rollouts. For Car Racing, Pong, Breakout, and Space Invaders, we collected 20 demonstration rollouts. Expert demonstrations were generated from scratch for Lunar Lander using DQN (Mnih et al., 2015), and collected from open-source pre-trained policies for Car Racing (Ha & Schmidhuber, 2018) as well as Humanoid and HalfCheetah (Dhariwal et al., 2017). The Humanoid demonstrations were generated by a stochastic expert policy, while the HalfCheetah demonstrations were generated by a deterministic expert policy; both experts were trained using TRPO.[6] We used two open-source implementations of GAIL: (Fu et al., 2017) for Lunar Lander, and (Dhariwal et al., 2017) for MuJoCo. We adapted the OpenAI Baselines implementation of GAIL to use soft Q-learning for Car Racing and Atari. Expert demonstrations were generated from scratch for Atari using DQN.

For Lunar Lander, we set $\lambda_{\text{samp}} = 10^{-6}$. For Car Racing, we set $\lambda_{\text{samp}} = 0.01$. For all other environments, we set $\lambda_{\text{samp}} = 1$.

---

[6]https://drive.google.com/drive/folders/1h3H4AY_ZBx08hz-Ct0Nxxus-V1melu1U

SQIL was not pre-trained in any of the experiments. GAIL was pre-trained using BC for HalfCheetah, but was not pre-trained in any other experiments.

In standard implementations of soft Q-learning and SAC, the agent's experience replay buffer typically has a fixed size, and once the buffer is full, old experiences are deleted to make room for new experiences. In SQIL, we never delete demonstration experiences from the replay buffer, but otherwise follow the standard implementation.

We use Adam (Kingma & Ba, 2014) to take the gradient step in line 4 of Algorithm 1.

The BC and GAIL performance metrics in Section 4.3 are taken from (Dhariwal et al., 2017).[7]

The GAIL and SQIL policies in Section 4.3 are set to be deterministic during the evaluation rollouts used to measure performance.

---

[7]https://github.com/openai/baselines/blob/master/baselines/gail/result/gail-result.md

