# OpenReview forum: "SQIL: Imitation Learning via Reinforcement Learning with Sparse Rewards"
_ICLR.cc/2020/Conference — Accept (Poster)_

### Official Review · AnonReviewer3 · 2019-10-22
**Official Blind Review #3**

**Rating:** 6

**Review:**

Summary
-------------
The authors propose SQUIL, an off-policy imitation learning (IL) algorithm which attempts to overcome the classic drift problems of behavioral cloning (BC). The idea is to reduce IL to a standard RL problem with a reward that incentivizes the agent to take expert actions in states observed in the demonstrations. The algorithm is tested on both image-based tasks (Atari) and continuous control tasks (Mujoco) and shown effective against GAIL and a simple supervised BC approach.

Comments
--------
Overcoming the limitations of BC algorithms is a relevant problem and this work presents a simple and interesting solution. The idea is to use any off-policy RL algorithm (assuming access to the true environment) with a reward that favors imitation. The paper is well-organized and easy to read. The experiments are also quite convincing and demonstrate that the proposed methodology is, in fact, more robust to distribution shift (i.e., less prone to overfitting the train set) than standard supervised BC and GAIL. Some comments/questions follow.

1. The considered settings (e.g., MDPs, IL/IRL problems) are never formalized in the paper (which directly starts by describing the method in Section 2). It would be good to add some preliminaries to allow even readers unfamiliar with this problem to easily go through the paper.

2. Though the approach solves one of the main problems of many BC/non-IRL algorithms, it still has some of the other limitations. For instance, the recovered policy is non-transferable to different dynamics, while a reward function is. Furthermore, it cannot recover optimal behavior if the expert is sub-optimal.

3. In many practical cases, the assumption that we can access the true environment is not reasonable. Suppose, for instance, that we have a dataset of driving demonstrations from a real car, and that is all we can obtain. Of course, we cannot start interacting with the real car to imitate the driver's behavior and we need to rely on simulators, which poses additional difficulties due to the inevitable approximations. Do the authors think that a batch RL algorithm could be adopted to solve this task (i.e., by solely using the expert data)? The ablation study in section 4 seems to answer this question negatively, at least for large distribution shifts.

4. Below equation (1), why "r does not depend on the state or action"? It should be R(s,a) = INDICATOR{(s,a) \in D_{demo}}, so it does depend on (s,a).

5. D_{demo} is missing from the inputs to Algorithm 1

6. I did not understand why, in Section 3.2 (above Equation 8), the authors mention that (7) squared is equivalent to the squared Bellman error of Equation 1. Where is r? It should be equivalent only from (s,a) \in D_samp, for which r=0, but for (s,a) \in D_demo we have r=1 and that term is missing in (7). Have I misunderstood something?

7. In Figure 1, why does GAIL perform better with shift than no-shift?

8. Why are some curves in Figure 3 (bottom) above the expert performance?

9. It would be interesting to show how the proposed method compares to IRL-based approaches for continuous states/actions such as [1] or [2]. These algorithms should be applicable to the considered domains (at least Mujoco ones).

[1] Finn, C., Levine, S., & Abbeel, P. (2016, June). Guided cost learning: Deep inverse optimal control via policy optimization. In International Conference on Machine Learning (pp. 49-58).
[2] Boularias, A., Kober, J., & Peters, J. (2011, June). Relative entropy inverse reinforcement learning. In Proceedings of the Fourteenth International Conference on Artificial Intelligence and Statistics (pp. 182-189).

**Experience Assessment:**

I have published one or two papers in this area.

**Review Assessment: Checking Correctness Of Derivations And Theory:**

I assessed the sensibility of the derivations and theory.

**Review Assessment: Checking Correctness Of Experiments:**

I assessed the sensibility of the experiments.

**Review Assessment: Thoroughness In Paper Reading:**

I read the paper at least twice and used my best judgement in assessing the paper.

---

> ### Author Response · Authors · 2019-11-11
> **Response to R3**
>
> Thank you for the thoughtful feedback. We agree with points 1, 2, 4, 5, and 9, and will update the paper according to the reviewer’s suggestions.
>
> Regarding point 6, we apologize for not defining the penalty term in RBC more clearly. The reviewer may have misunderstood our notation for the penalty term. In Equation 8 and the paragraph above Equation 8, r=0 for both (s,a) \in D_samp and (s,a) \in D_demo. In RBC, we use r=0 for regularization via the penalty term, and use the BC loss to encourage imitation. In SQIL, we use r=0 for regularization, and use r=1 in the demonstrations to encourage imitation.
>
> Regarding point 7, the performance metrics for a given method with shift vs. no shift are not comparable, since the agent starts in different initial states. Only the metrics for different methods within the same column are comparable.
>
> Regarding point 8, performance on these tasks may depend on the stochasticity of the policy. For some of the tasks, we used demonstrations from a stochastic expert, but evaluated the imitation policy by selecting actions deterministically (see details in Section A.3 in the appendix).

---

> > ### Comment · AnonReviewer3 · 2019-11-15
> > **Post-rebuttal comments**
> >
> > Thank you for the detailed response and for clarifying my doubts.  I confirm my initial view and vote for acceptance. I also recommend the authors to update the paper according to the suggestions of all reviewers.

---

### Official Review · AnonReviewer1 · 2019-10-24
**Official Blind Review #1**

**Rating:** 6

**Review:**

This paper proposes an imitation learning approach via reinforcement learning. The imitation learning problem is transformed into an RL problem with a reward of +1 for matching an expert's action in a state and a reward of 0 for failing to do so. This encourages the agent to return to "known" states from out-of-distribution states and alleviates the problem of compounding errors. The authors derive an interpretation of their approach as regularized behavior cloning. Furthermore, they empirically evaluate their approach on a set of imitation learning problems, showing strong performance. The authors also stress the easy implementability of their approach within standard RL methods.

I think this is a good paper which shows strong empirical results based on simple but effective idea. I would welcome an extended discussion of certain aspects of the experiments though -- for instance the trends in Figure 3. Furthermore, I think the huge gap in performance between SQIL and RBC warrants a more detailed discussion/analysis. Also the papers by Sasaki et al. (which was accepted at last year's ICLR) [and (maybe) the paper by Wang et al.] deserve to be discussed earlier in the paper (introduction?) and make it to the experimental results.

The paper could be improved considering the following minor suggestions:
* Introducing the important bits of soft-Q learning formally.
* Defining demonstrations formally (not only in the algorithm).
* Showing the necessity to balance demonstrations and new experiences in the algorithm.
* Removing implementation details from Section 2 and adding them to the experiments.
* There is a bunch of repetitive paragraphs, that could be cleaned up.
* Equation (2) and (5) are the same. Use $\pi^*$ and $Q^*$ in (2)?
* I disagree with paragraph following equation (6). If we phrase BC as an optimization problem over Q then Q should satisfy consistency. Otherwise, we are not optimizing over Q but some other object.
* Please specify the initial state distribution.

**Experience Assessment:**

I have published one or two papers in this area.

**Review Assessment: Checking Correctness Of Derivations And Theory:**

I assessed the sensibility of the derivations and theory.

**Review Assessment: Checking Correctness Of Experiments:**

I assessed the sensibility of the experiments.

**Review Assessment: Thoroughness In Paper Reading:**

I read the paper at least twice and used my best judgement in assessing the paper.

---

> ### Author Response · Authors · 2019-11-11
> **Response to R1**
>
> Thank you for the thoughtful feedback. We will add a more detailed discussion of the results in Figure 3 and the gap between SQIL and RBC in the final paper. We will also move the discussion of Sasaki et al. and Wang et al. to the introduction, and address the reviewer’s remaining suggestions.

---

### Official Review · AnonReviewer2 · 2019-10-25
**Official Blind Review #2**

**Rating:** 8

**Review:**

This paper proposes a simple method for imitation learning that is competitive with GAIL.  The approach, Soft Q Imitation Learning (SQIL), utilizes Soft Q Learning, and defines high rewards as faithfully following the demonstrations and low rewards as deviating from them.  Because SQIL is off-policy, it can utilize replay buffers to accelerate sample efficiency.  One can also interpret SQIL as a regularized version of behavioral cloning.

The authors really play up the "surprising" connection of SQIL to a regularized behavioral cloning.  But this isn't really that surprising in the general sense (although I applaud the authors for rigorously defining the connection).  SQIL is basically adding "pseudo-data" and the idea of using pseudo-counts as regularization has been around for a long time.   I think it's a stretch to hype up the "surprise" factor so much, and that it does a disservice to the paper overall.

The experiments are sound, showing competitive performance to GAIL, and also having a nice ablation study.

I wonder if there was a quantitative way to demonstrate that SQIL is "simpler" than GAIL.  It's certainly conceptually easier.  Is the implementation in any concrete sense easier as well?  Or sensitivity to hyperparameters or initial conditions?

**Experience Assessment:**

I have published one or two papers in this area.

**Review Assessment: Checking Correctness Of Derivations And Theory:**

I carefully checked the derivations and theory.

**Review Assessment: Checking Correctness Of Experiments:**

I carefully checked the experiments.

**Review Assessment: Thoroughness In Paper Reading:**

I read the paper thoroughly.

---

> ### Author Response · Authors · 2019-11-11
> **Response to R2**
>
> Thank you for the thoughtful feedback. We will adjust the language used to describe the connection between SQIL and RBC in the final paper.
>
> Since SQIL does not require a discriminator, it uses approximately half the number of model parameters as GAIL. SQIL also uses fewer hyperparameters; e.g., it does not require tuning the number of discriminator gradient steps per RL gradient step.

---

### Decision · Program_Chairs · 2019-12-19

**Decision:**

Accept (Poster)

**Comment:**

The authors present a simple alternative to adversarial imitation learning methods like GAIL that is potentially less brittle, and can skip learning a reward function, instead learning an imitation policy directly.  Their method has a close relationship with behavioral cloning, but overcomes some of the disadvantages of BC by encouraging the agent via reward to return to demonstration states if it goes out of distribution.  The reviewers agree that overcoming the difficulties of both BC and adversarial imitation is an important contribution.  Additionally, the authors reasonably addressed the majority of the minor concerns that the reviewers had.  Therefore, I recommend for this paper to be accepted.